# High-Permittivity Silicone Composites with Different Polarization Titanates for Electric Field Modification

**DOI:** 10.3390/polym17070986

**Published:** 2025-04-04

**Authors:** Evgeniy Radzivilov, Ilya Zotov, Maria Vikulova, Alexey Tsyganov, Ivan Artyukhov, Denis Artyukhov, Alexander Gorokhovsky, Artem Yudin, Nikolay Gorshkov

**Affiliations:** 1School of Advanced Manufacturing Technologies, Tomsk Polytechnic University, 30 Lenina Avenue, 634050 Tomsk, Russia; 2Department of Chemistry and Technology of Materials, Yuri Gagarin State Technical University of Saratov, 77 Politechnicheskaya Street, 410054 Saratov, Russia; 3Department of Power and Electrical Engineering, Yuri Gagarin State Technical University of Saratov, 77 Polytechnicheskaya Street, 410054 Saratov, Russia

**Keywords:** barium titanate, calcium copper titanate, hollandite, silicone, polymer composite, permittivity, dielectric loss

## Abstract

Polymer-matrix composites with ceramic fillers have various applications, one of which is the modification of the electric field. For this purpose, in this work, high-permittivity silicone composites with different polarization titanates were produced by mechanical mixing. The ceramic fillers chosen were CaCu_3_Ti_4_O_12_, K_x_Fe_y_Ti_8−y_O_16_, and BaTiO_3_ powders with high permittivity values and uniformly distributed in the polymer volume. Ceramic powders were studied by X-ray phase analysis and scanning electron microscopy methods. The proportion of ceramic powder was 25 wt.%. In parallel, composites were prepared with the addition of 25 wt.% glycerin. The functional properties of silicone composites were studied using the following parameters: the electrical strength and permittivity. The addition of all types of ceramic fillers, both together and without glycerin, led to a decrease in electrical strength (below 15 kV·mm^−1^); the exception is the sample with the CCTO without glycerin (about 28 kV·mm^−1^). The permittivity and the dielectric loss tangent of the composites increased as a result of the addition of fillers, especially noticeable in combination with glycerol in the low-frequency region. The obtained results are in good agreement with the literature data and can be used in the field of insulation in a high-permittivity layer to equalize equipotential fields.

## 1. Introduction

In recent decades, there has been a significant increase in attention to the application of high-voltage pulse technology for well drilling, mineral disintegration, concrete and reinforced concrete demolition, electronic device recycling, high-voltage lines in power supply systems for multigenerator microwave installations, and other similar tasks [1,2]. Effective insulation of high-voltage equipment is a complex task, since it is necessary to take into account the change in the direction of the electric field lines in places where the integrity of the main insulation with high electrical strength and low permittivity is broken [3,4,5]. A similar problem arises when designing the electrode unit of electric pulse installations, for example, for the destruction of concrete, since it is necessary to direct the equipotential fields as much as possible in the same direction as the electrodes. One of the solutions for aligning high-tension electric field lines of high-voltage equipment is the refraction method of alignment of the electric field [6,7]. The essence of the method is to use an external insulating shell with a permittivity significantly exceeding the permittivity of the main insulator.

To solve this problem, a well-known, effective and easily implemented approach is the creation of polymer dielectric composites, which are polymer dielectrics with inorganic fillers uniformly distributed in their volume [8,9]. As fillers, various oxide and carbon materials are usually used [10]. The positive role of the optimal content of dicumyl peroxide in improving the dielectric characteristics of direct current and the mechanical properties of cross-linked polyethylene for use in cable insulation has also been proven [11]. It is shown that chemical modification of the ethylene-propylene-diene polymer matrix with glyceryl monooleate molecules is implemented to improve electrical conductivity and achieve nonlinear conductivity, which is proposed to be used to simultaneously achieve high compliance of electrical characteristics between reinforced and basic insulation in cable lugs and maintain sufficient dielectric breakdown strength [3].

Silicones meet the requirements for mechanical and electrical properties of insulating materials, taking into account that silicones do not form conductive carbon particles during electrical breakdown or sparking [12]. The dielectric constant of silicones varies from 2.8 to 3.3 [13,14]. They are characterized by flexibility and ease of molding into products of various shapes and sizes and can therefore be considered as a polymer base for a composite.

To increase the permittivity of the copolymer, Zhang et al. proposed adding special components with high permittivity, such as copper phthalocyanine [15]. Recently, titanium dioxide (TiO_2_) has been investigated as a ceramic filler for soft polydimethylsiloxane matrices to develop an elastomeric composite with improved electromechanical properties [16].

There are many studies devoted to the development of insulating materials with increased permittivity [13,17,18,19,20]. However, these materials do not meet our requirements due to low permittivity at frequencies above 1 kHz (the permittivity value does not exceed 32 at 1 kHz and decreases with increasing frequency).

Also, the best results were achieved in [21]; namely, the permittivity of polyimide doped with modified silicon dioxide with titanium carbide nanoparticles is 52 at a frequency of 1 kHz (with increasing frequency, the permittivity decreases significantly).

In this study, the choice of fillers in the form of complex oxides of the compositions CaCu_3_Ti_4_O_12_ (CCTO), K_x_Fe_y_Ti_8−y_O_16_ (KFTO), and BaTiO_3_ (BTO) is due to differences in their structure, polarization mechanism and dielectric properties. Thus, CCTO has an electrical microstructure consisting mainly of semiconducting grains and insulating grain boundaries. This microstructure leads to the development of Schottky barriers at grain boundaries, thereby creating the internal barrier layer capacitance (IBLC) mechanism, which is used to explain the giant value of the permittivity [22,23]. In the case of KFTO, the increase in permittivity occurs due to the increase in polarization associated with the variable valence of the structure-forming elements, namely titanium and iron [24]. BTO in turn has a domain-dependent electronic structure [25]. Thus, it is of particular interest to trace how the individual characteristics of ceramic fillers will influence the functional properties of the polymer matrix.

Our previous works [26,27,28,29,30] have shown the successful introduction and distribution of CCTO and KFTO in PVDF, PMMA and PTFE polymer matrices to obtain composite materials; therefore, the approach used can be effectively adapted to create composites based on other polymers. Therefore, the selection of filler for silicone composites with high permittivity and good flexibility remains a relevant and multifactorial task. The silicone polymer matrix has a Shore hardness of around 30, while the previously studied PVDF, PMMA and PTFE are over 100 on scale A and 65–82, 90–99 and 50–65 on scale D, respectively.

The aim of this work is to create silicone polymer-matrix composites with an expected increase in dielectric permittivity based on silicone and fillers in the form of titanates with different types of polarization.

The effectiveness of ceramic titanate powders in creating polymer-matrix composites with an optimal combination of dielectric properties has already been demonstrated in systems [24,26,27,28,29,30]. Glycerin is widely used as a plasticizer for polymers, including silicones, and will preserve the elasticity of composites after the addition of fillers, which together opens up wide possibilities for the development of new polymer-matrix composites with given dielectric properties [31,32,33].

The aim of this work is to synthesize and study composites based on silicones, titanates with different types of polarization and glycerin.

## 2. Materials and Methods

### 2.1. Raw Materials

A high-strength, non-shrink silicone compound Pentelast-750 of grade A (New Composite, Saint Petersburg, Russia) was used as a silicone matrix. Pentelast-750 is a two-component compound consisting of component A and component B, which, when mixed, hardens at room temperature according to the following reaction (Figure 1):

The silicone used has the following characteristics (Table 1):

As the addition of ceramic powders to polymer matrices reduces the elasticity of composites, and as the filler content increases, the viscosity decreases and the homogenization efficiency decreases, the use of plasticizers may be a reasonable solution. Glycerin (Reakhim, Moscow, Russia) is used as a plasticizer.

### 2.2. Synthesis of Fillers

CaCu_3_Ti_4_O_12_ (CCTO) powder was synthesized by solid-state reaction using the methodology in [26]. The raw materials were CaCO_3_ (99%, Reakhim, Moscow, Russia), CuO (99%, Reakhim, Moscow, Russia), and TiO_2_ (anatase, 99% Component reaktiv, Moscow, Russia). Mixing and mechanochemical activation of the powders taken in a stoichiometric ratio was carried out using a Fritsch Pulverisette 6 planetary mill with a rotation speed of 500 rpm for 2 h. The chemical reaction in an electric furnace was under the temperature of 900 °C for 3 h. The synthesized CCTO powder was ground in an alcohol medium using the Fritsch Pulverisette 6 planetary mill with a rotation speed of 600 rpm for 1 h.

K_x_Fe_y_Ti_8−y_O_16_ (KFTO) (x = 1.4–1.8, y = 1.2–1.6) powder was synthesized by the Pechini sol–gel method using the methodology in [27,28]. The raw materials were C_2_H_6_O_2_ (98.5%, Aricon, Moskow, Russia), C_6_H_8_O_7_ (99.5%, Aricon, Moskow, Russia), KNO_3_ (98%, Buyskiy himicheskiy zavod, Buy, Russia), Fe(NO_3_)_3_ 9H_2_O (98%, Buyskiy himicheskiy zavod, Buy, Russia), C_16_H_36_O_4_Ti (99%, Acros Organics, Geel, Belgium), HNO_3_ (65%, Buyskiy himicheskiy zavod, Buy, Russia), and NH_4_OH (25%, Aricon, Moskow, Russia). The sol was formed by mixing the solutions of KNO_3_, Fe(NO_3_)_3_ 9H_2_O and C_16_H_36_O_4_Ti in stoichiometric quantities and adding HNO_3_, C_2_H_6_O_2_ and C_6_H_8_O_7_ as well as NH_4_OH until pH = 8. The gel was obtained after evaporation of the solvent at 240 °C. For crystallization, the gel was calcined at 900 °C for 1 h.

BaTiO_3_ (BTO) powder was synthesized by the oxalate method [34,35]. The raw materials were 50% aqueous solution of TiCl_4_ (Promhim, Sverdlovsk, Russia), BaCl_2_·2H_2_O (Adelit, Ufa, Russia), and H_2_C_2_O_4_·2H_2_O (Chemservice, Tula, Russia). A 50% solution of TiCl_4_ was added to a supersaturated solution of H_2_C_2_O_4_·2H_2_O. The reaction mixture was stirred for 10 min, during which the oxalic acid dissolved completely and the mixture heated naturally; as a result, a transparent solution was obtained. An aqueous solution of BaCl_2_·2H_2_O was used as a precipitant at a drop rate of 80–100 mL/min. After mixing all the solutions, the mixture was stirred for 1 h. The resulting white precipitate was washed using distilled water to pH = 3.5–4.0 and by grinding in an organic medium (80 vol.% H_2_O, 15 vol.% aminoethanol and 5 vol.% propanol) at 300 rpm for 30 min to pH = 9.0–10.0, followed by washing with distilled water to pH = 7.0. The precipitate was dried at 150 °C and calcined at 1100 °C for 2 h (a heating rate of 200 °C/h).

### 2.3. Production of Composites

Addition of ceramic fillers into two-component silicone M750 was carried out at room temperature by mechanically mixing ceramic powders with component A, followed by the addition of a polymerization activator (component B). To achieve this, a dose of component A was measured into the container, to which, with constant stirring, a portion of the ceramic filler was gradually added and mixed together. After BaTiO_3_/KFTO/CCTO powder was evenly distributed in the volume of component A, the required portion of component B was added. The proportion of ceramic powder was 25 wt.%. The mixture was subjected to a vacuum process, casting into a cylindrical form to give it the required shape for subsequent studies followed by vacuuming and drying (Figure 2).

The addition of ceramic fillers in combination with glycerin into silicone was carried out using a similar technique. In addition, 25 wt.% of glycerin was added.

### 2.4. Characterization of Fillers

Characterization of ceramic powders was carried out using X-ray phase analysis (X-ray diffractometry using Cu(Kα) radiation (λ = 0.15412 nm) (Thermo Scientific ARL X’TRA, Ecublens, Switzerland)) and scanning electron microscopy methods (ASPEX Explorer Scanning Electron Microscope (ASPEX, Framingham, MA, USA)).

### 2.5. Study of Composites

The electrical strength and permittivity of the composites were measured.

To determine permittivity, the capacitance (*C*) was measured at three frequencies (1 kHz, 100 kHz, 1 MHz) at room temperature using the E7-20 Immittance Meter (OJSC “MNIPI”, Republic of Belarus, Minsk). For this, the samples were placed in a measuring cell, which was a simple flat capacitor with round electrodes with a diameter of 48 mm. The permittivity was calculated from the formula for the capacitance of a flat capacitor [5]:(1)$$C=\frac{\varepsilon \varepsilon_{0}S}{d}$$(2)$$\varepsilon =\frac{Cd}{\varepsilon_{0}S}$$
where *C* is the capacitance; *Ɛ* is the permittivity; *Ɛ*_0_ is the electric constant (8.854 · 10^−12^ F/m); *S* is the electrode area; *d* is the distance between electrodes (thickness of dielectric material).

The breakdown voltage was measured at room temperature using an AIM-90 device (“Mosrentgen”, Mosrentgen USSR.). The samples were placed in a measuring cell, which was a container with electrodes located at a fixed distance from each other. Then, the electrical strength of the material can be calculated:(3)$$E_{b}=\frac{U_{b}}{d}$$
where *E_b_* is the electrical strength; *U_b_* is the breakdown voltage; *d* is the thickness of dielectric material.

To determine the interaction of fillers with silicone M750, samples were studied using the Scanning Electron Microscope S-3400N device (Hitachi, Japan, Tokyo).

## 3. Results and Discussion

The synthesized ceramic materials were studied by X-ray phase analysis to confirm the structure and scanning electron microscopy to estimate the particle size before being added into the polymer matrix. The results obtained are shown in Figure 3 and Figure 4.

All synthesized ceramic powders are described by X-ray diffraction patterns; they are characterized by a crystalline structure and are monophasic. According to XRD data, BTO powder has a perovskite tetragonal crystal structure (space group P4mm). Observed diffraction peaks at 2θ = 22°, 32°, 39°, 45°, 51°, and 56° correspond to the planes of (100), (110), (111), (200), (210), and (211), respectively. The X-ray diffraction pattern of KFTO powder includes the diffraction peaks at 2θ = 12°, 18°, 25°, 28°, 35°, 36°, 39°, 40°, 46°, and 48°, related to the priderite phase with a tetragonal hollandite-like structure (space group l4/m). The XRD pattern of CCTO powder also has several characteristic peaks at 2θ = 29°, 34°, 38°, 42°, 46°, 49°, and 62°, confirming the formation of a body-centered cubic perovskite structure (space group Im3). It should be noted that in all cases there are no secondary crystalline phases, which indicates the successful synthesis of materials by the selected methods.

An analysis of the morphology and particle size is carried out using scanning electron microscopy. Particles of synthesized ceramic materials do not have a specific shape but have a uniform distribution. BTO particles tend to be spherical in shape; their diameter is about 1 μm. KFTO is characterized by more elongated particles with a length of no more than 1 µm and a width of about 0.2 µm. CCTO particles with an average diameter of 0.5 µm form fairly large agglomerates of several µm. The particle size depends on the method of ceramic powder synthesis, and the method of further processing and has a significant impact on the uniformity of distribution in the polymer matrix and the change in its functional properties.

Visual analysis of the produced composites at the breakdown site (Figure 5) shows the presence of a significant amount of air inclusions, which affect the studied functional properties. In the case of composites with KFTO and CCTO fillers, these spherical inclusions are quite large. In comparison with them, the use of BTO as a ceramic filler reduces the presence and size of air inclusions.

The presence of air pores is further confirmed by scanning electron microscopy data (Figure 6). Their greatest intensity is observed in the sample of silicone/CCTO/glycerin. Electron micrographs can be used to assess the uniformity of distribution of all ceramic fillers in the polymer matrix, which appear as lighter inclusions in the images.

The study of the electrical strength of silicone before and after the addition of ceramic fillers shows the following results: the addition of all types of ceramic fillers, both together and without glycerin, leads to a decrease in electrical strength. The exception is the sample with the CCTO. When producing composites using glycerin, the decreasing in electrical strength is more significant (more than 3 times) (Figure 7).

The influence of fillers of different compositions, structures and combinations on the dielectric properties of silicone in the frequency range of 10^2^–10^6^ Hz were studied (Figure 8 and Figure 9).

The permittivity of silicone in the range studied is a frequency-independent and maintains a value at the level of 2.6 units. The addition of ceramic fillers (BTO, KFTO, CCTO) in an independent form does not change the behavior of frequency dependence but leads to an increase in the permittivity up about to 5 units. However, the addition of the same ceramic fillers in combination with glycerin causes a simultaneous increase in the permittivity in the entire frequency range studied, especially in the low-frequency region (10^2^–10^3^ Hz). In this case, it is necessary to note the influence of the type of ceramic filler in the low-frequency region. At *f* = 10^2^–10^3^ Hz, the highest permittivity is demonstrated by silicone with KFTO as a filler, with the lowest by silicone with BTO. The described behavior is normal for such composites [36,37,38].

By analogy with the permittivity, the addition of fillers leads to an increase in the dielectric loss tangent relative to pure silicone, characterized by tanδ = 0.01 (Figure 9). Ceramic fillers cause almost the same and less significant increase in tanδ up to about 0.03. In combination with glycerin, this increase is more significant, especially in the low-frequency range, up to 10 units.

The literature data on dielectric properties of polymer-matrix composites based on silicones using various ceramic fillers are given in Table 2. It should be noted that the concentration range of the titanate fillers is from 2 to 25 wt.%, with the addition of plasticizers and conductive additives such as CNTs and MXenes. Titanates and silicones have a different nature of polarization processes and conductivity and are therefore characterized by the occurrence of Maxwell–Wagner polarization at interfaces. Silicone has a density of 1.08 g/cm^3^; barium titanate has a density of 5.8 g/cm^3^; KFTO has a density of 3.85 g/cm^3^; and CCTO has a density of 4.7 g/cm^3^. The volumetric contents of 25 wt.% titanates in the composites are then estimated to be 5.85 vol.% for BTO, 8.54 vol.% for KFTO and 7.12 vol.% for CCTO. It is worth noting that the permittivity values for silicone composites with titanates of different polarization types are close in the case without plasticizer (glycerin). In comparison with literature data, the permittivity of the obtained composites is in the same range as similar systems without plasticizers and conductive additives. The addition of glycerin with intrinsic permittivity of about 43 in an amount of 25 wt.% (about 22 vol.%) increases the permittivity for the composite with BTO to 9.21, for KFTO to 54.41 and for CCTO to 16.18. Plasticizers are known to improve the tribological properties of composites and to promote better filler distribution in composites. However, in the systems investigated, in addition to the increase in permittivity values, which can be explained by improved distribution of the filler and the contribution of glycerin, a change in the form of their frequency dependence is observed. In composites with glycerin at low frequencies (100–500 Hz), the permittivity and the dielectric loss tangent for the titanate sample KFTO, characterized by ionic conductivity, increase significantly with decreasing frequency. For the CCTO filler, the effect of electrode polarizations is less pronounced than for the KFTO sample.

Table 3 shows the Shore hardness values for the investigated silicone-matrix composites. As expected, the Shore hardness increases from 30 to 38, 42 and 47 after the introduction of BTO, CCTO and KFTO ceramic powders, respectively.

The addition of plasticizer in the form of glycerol at 15% significantly reduces the Shore hardness to 11, 16 and 17 for composites filled with BTO, CCTO and KFTO ceramic powders, respectively.

The use of capacitive (or high-dielectric) stress relief offers distinct advantages in design and application. A simple extruded tube replaces the complex rubber molded cone. By eliminating the inhomogeneity of the electric field at the end of the insulating shield (Figure 10), the addition of a high-permittivity material from the end of the shield bends the electric field in the direction of the cable. Figure 10 shows a clear field distribution of the cable terminal before and after the use of high-k rubber–matrix materials. The magnitude of this refraction is determined by the angle of incidence of the flux lines passing from one dielectric to the other and the ratio of the permittivity of these materials, and is based on the formula shown in Figure 10.

The type of polarizations therefore has a significant effect on the frequency dependence of silicone composites. The possibility of using silicone composites as an insulator in a high-permittivity layer to equalize equipotential fields can be considered. For example, for networks with a frequency of 50 Hz, ionic polarization fillers can be effective for the range 500 Hz–10 kHz–IBLC, for high frequencies with domain polarizations.

For example, high-k elastic silicone-matrix composites, which could balance the electric field distribution of cable terminals to prevent cable failure, could be used as potential functional materials for cable accessories in electrical engineering.

## 4. Conclusions

This study investigated the influence of the addition of ceramic fillers with different types of polarization with and without plasticizer on the electrical strength and dielectric properties of silicone composites. For this purpose, the following ceramic fillers were chosen: CaCu_3_Ti_4_O_12_ with IBLC polarization, K_x_Fe_y_Ti_8−y_O_16_ with ion polarization, and BaTiO_3_ with domain polarization. Glycerin was used as a plasticizer. All monophase ceramic powders are characterized by an undefined morphology and an average size of about 1 µm, which contributes to their uniform distribution in the polymer.

The addition of all types of ceramic fillers, both together and without glycerin, leads to a decrease in electrical strength from ~18 kV·mm^−1^ for pure silicone to a minimum value of ~4 kV·mm^−1^ for the silicone/KFTO/glycerin composite. An exception is a sample of silicone/CCTO characterized by an increased value of electrical strength of about 28 kV·mm^−1^.

The dielectric properties of silicone composites in the frequency range of 10^2^–10^6^ Hz were studied. The permittivity and dielectric loss tangent of pure silicone are frequency-independent quantities (ε = 2.6, tanδ = 0.01). The addition of ceramic powders without glycerin does not change the frequency behavior but leads to an increase in both ε up to 5 and tanδ up to 0.02–0.03. The addition of glycerin into the system causes a further increase in the dielectric parameters, especially in the low-frequency region.

Therefore, the type of polarizations has a significant effect on the frequency dependence of silicone composites. The possibility of using silicone composites as an insulator in a high-permittivity layer to equalize equipotential fields can be considered.

## Figures and Tables

**Figure 1 polymers-17-00986-f001:**
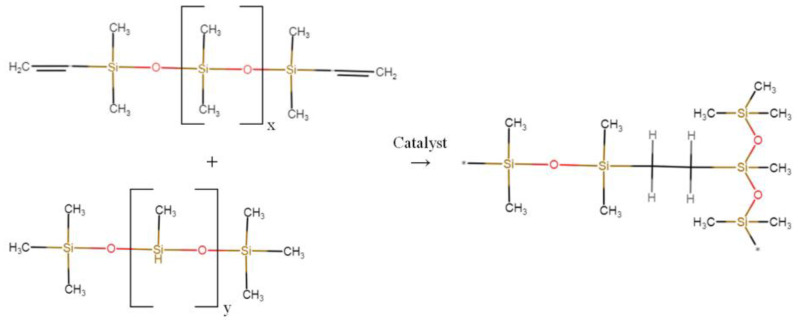
Polymerization reaction of Pentelast-750 silicone.

**Figure 2 polymers-17-00986-f002:**
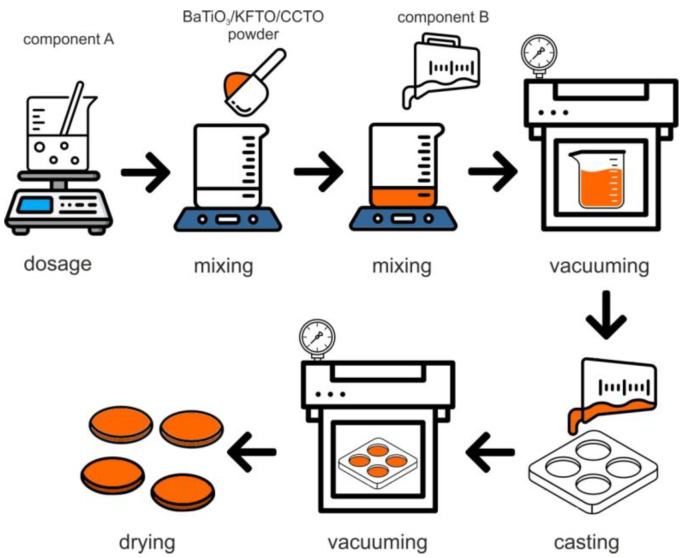
Scheme of composite production.

**Figure 3 polymers-17-00986-f003:**
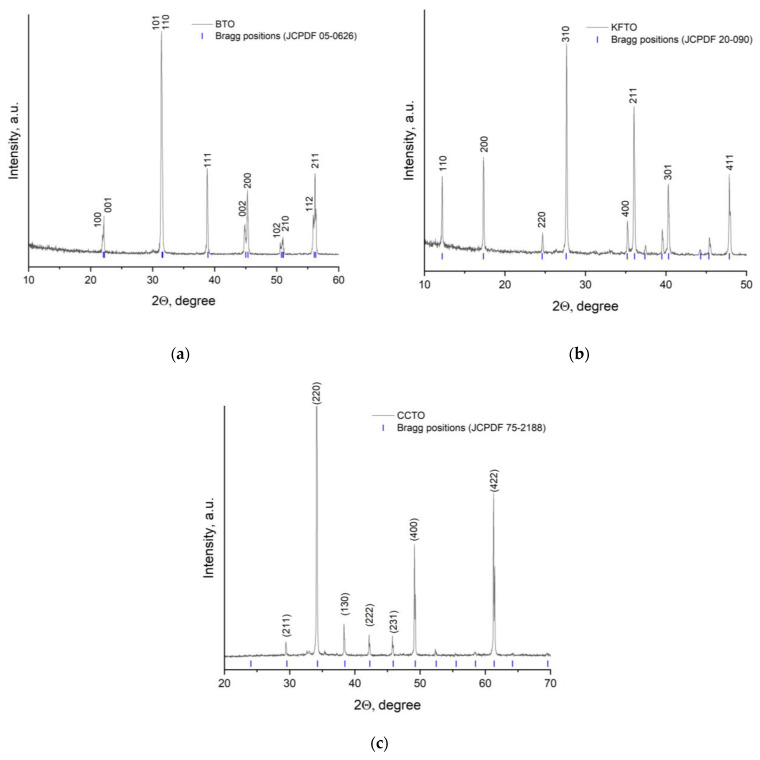
XRD patterns of (**a**) BTO, (**b**) KFTO and (**c**) CCTO powders.

**Figure 4 polymers-17-00986-f004:**
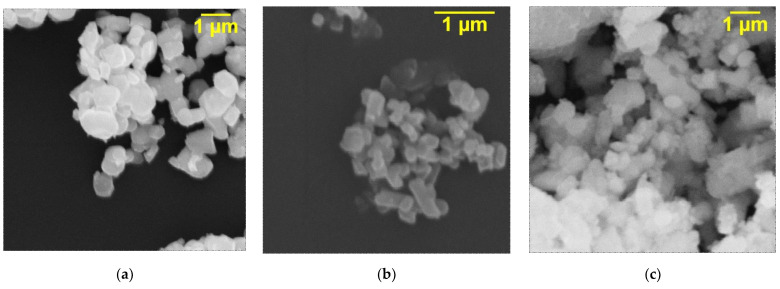
SEM images of (**a**) BTO, (**b**) KFTO and (**c**) CCTO powders.

**Figure 5 polymers-17-00986-f005:**
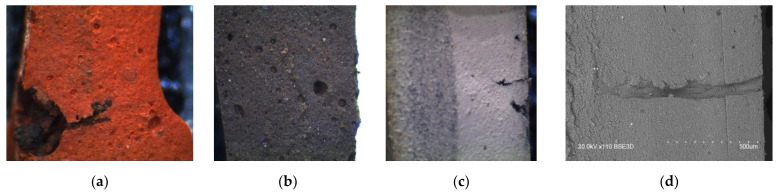
Visual analysis of the produced samples at the breakdown site: (**a**) silicone/KFTO/glycerin; (**b**) silicone/CCTO/glycerin; (**c**,**d**) silicone/BTO/glycerin.

**Figure 6 polymers-17-00986-f006:**
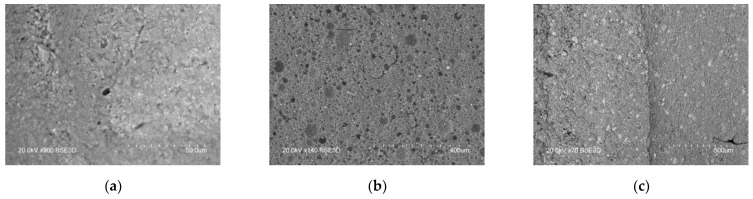
Scanning electron micrographs of the produced samples: (**a**) silicone/KFTO/glycerin; (**b**) silicone/CCTO/glycerin; (**c**) silicone/BTO/glycerin.

**Figure 7 polymers-17-00986-f007:**
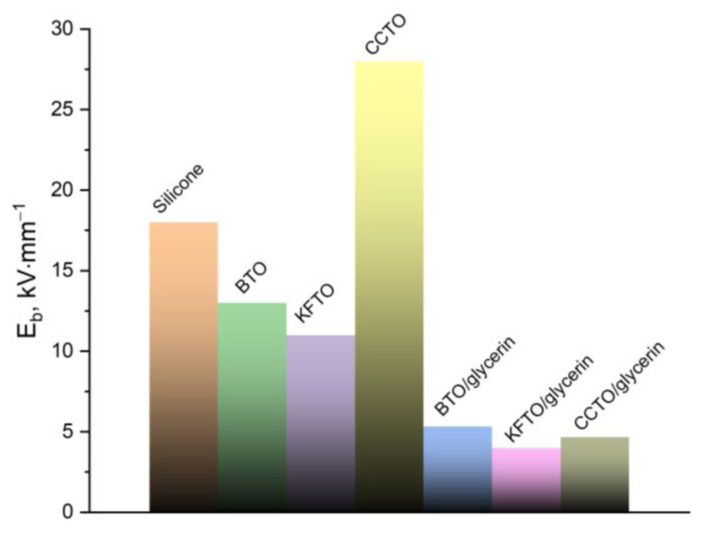
The electrical strength of silicone before and after filler addition.

**Figure 8 polymers-17-00986-f008:**
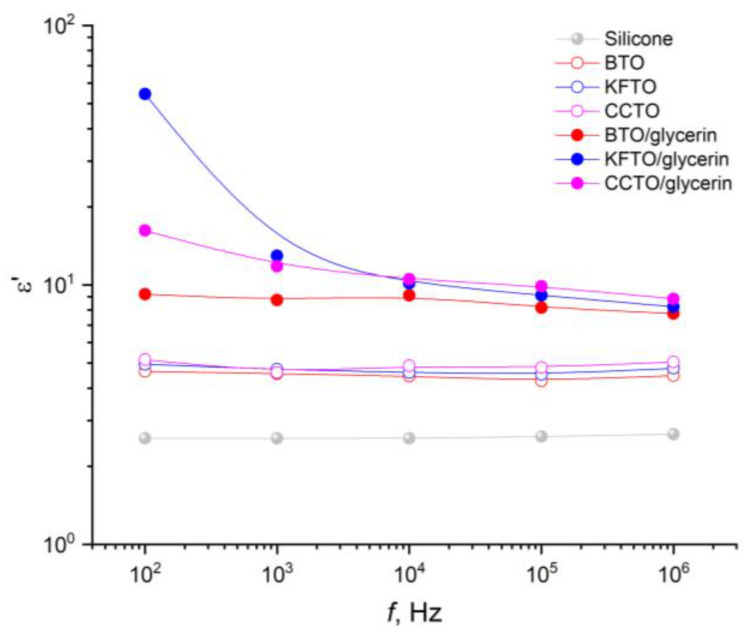
Frequency dependences of permittivity of silicone before and after filler addition.

**Figure 9 polymers-17-00986-f009:**
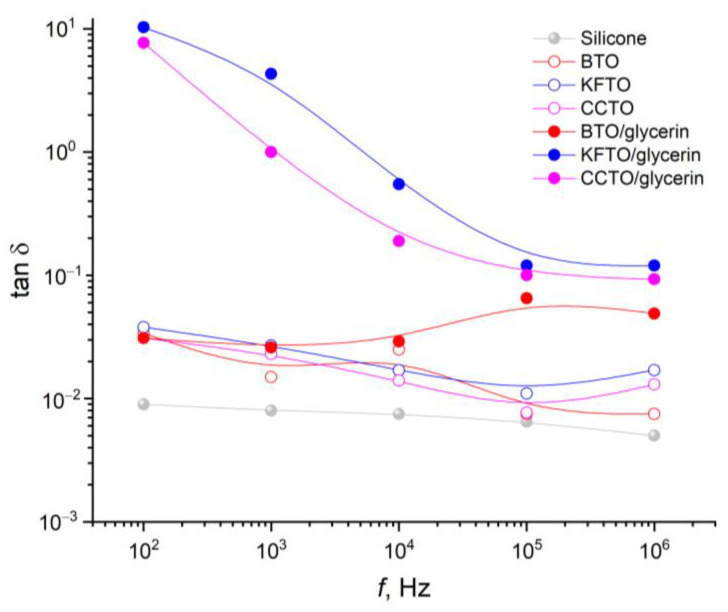
Frequency dependences of the dielectric loss tangent of silicone before and after filler addition.

**Figure 10 polymers-17-00986-f010:**
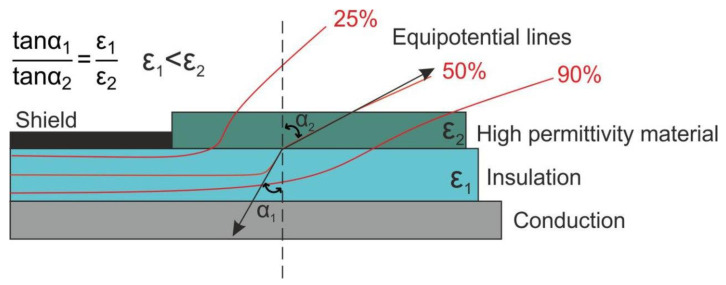
Electric field modified by stress relief with high permittivity.

**Table 1 polymers-17-00986-t001:** Characteristics of Pentelast-750 silicone.

Parameter	Value
Viscosity at 20 degrees, SPz	(8–20)·10^3^
Viability, h, not less than	1
Relative elongation at break,%, not less than	400
Tensile strength, MPa, not less than	3.0
Shore Hardness (Scale A)	35

**Table 2 polymers-17-00986-t002:** Dielectric properties of polymer-matrix composites based on silicones.

Silicone Composite	Filler	Filler Content (wt.%)	Polarization Type of Filler	Dielectric Constant (at Range 100 Hz–100 kHz)	Loss Tangent (at Range 100 Hz–100 kHz)
polydimethylsiloxane-α,ω-diol (M_w_ = 642,000 g mol^−1^)/30% SiO_2_/BTO [39]	BTO	15	Domain polarization	4.15–4.20	0.0150–0.0170
polydimethylsiloxane-α,ω-diol (M_w_ = 650,000 g mol^−1^)/15% PLURONIC L-31/BTO [40]	BTO	15		5.10–4.00	0.0130–0.1300
Commercially available silicone rubber NE 5140/BTO [41]	BTO	11		4.10–1.50	0.1400–0.4700
Silicone rubber/Ba_0.6_Sr_0.4_TiO_3_ [42]	Ba_0.6_Sr_0.4_TiO_3_	20		4.05–4.00	0.0300–0.0450
Commercially available silicone rubber Ecoflex 00–30 [12]	BTO	20		4.60–4.38	0.0180–0.0070
Commercially available silicone rubber M750/BTO [this work]	BTO	25		4.95–4.47	0.0340–0.0080
Commercially available silicone rubber M750/glycerin/BTO [this work]	BTO	25		9.21–7.75	0.0310–0.0650
Commercially methylvinyl silicone rubber type 110–2/2% CNTs/CaCu_3_Ti_4_O_12_/10% Benzoyl peroxide [43]	CCTO	10	IBLC	5.7	0.0012
Commercial methylvinyl silicone rubber type 110–2/CaCu_3_Ti_4_O_12_/1.2% Ti_3_C_2_T_x_ MXene [44]	CCTO	12		7	0.0016
Commercially available silicone rubber M750/CCTO [this work]	CCTO	25		5.14–5.04	0.0340–0.0075
Commercially available silicone rubber M750/glycerin/CCTO [this work]	CCTO	25		16.18–8.84	0.0310–0.0490
Commercially available silicone rubber M750/KFTO [this work]	KFTO	25	Ion polarization	4.94–4.77	0.0380–0.0110
Commercially available silicone rubber M750/glycerin/KFTO [this work]	KFTO	25		54.41–8.23	10.3–0.1200

**Table 3 polymers-17-00986-t003:** Shore hardness (scale A) of polymer-matrix composites based on silicones.

Silicone Composite	Filler	Shore Hardness
Commercially available silicone rubber M750/BTO	BTO	38
Commercially available silicone rubber M750/glycerin/BTO	BTO	11
Commercially available silicone rubber M750/CCTO	CCTO	42
Commercially available silicone rubber M750/glycerin/CCTO	CCTO	16
Commercially available silicone rubber M750/KFTO	KFTO	47
Commercially available silicone rubber M750/glycerin/KFTO	KFTO	17

## Data Availability

Data are contained within the article.

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
