# Peer review of "High-Permittivity Silicone Composites with Different Polarization Titanates for Electric Field Modification"

_polymers, 2025, doi:10.3390/polym17070986_

Round 1
Reviewer 1 Report
Comments and Suggestions for Authors
I suggest rewriting the abstract in a precise manner.
The title " High Permittivity Silicone Composites with Different Polarization Titanates for Electric Field Modification" clearly says high permittivity composites. Can you mention the frequency?
How to modify the electric field to get high permittivity?
The CaCO3 was used for synthesis. After reaction how can you confirm the absence of carbon dioxide?
Fig.1 is giving good graphical information for preparation. Provide some more discussion for this.
For equation 1 & 2, give references.
Fig.2 contains low resolution and i suggest comparing with JCPDS.
Determine the particle size for the samples using SEM.
Explain the reason for obtaining the shape of particles.
Fig.7 is for low frequencies. Are there any applications using this.
What is the difference between loss tangent and imaginary part of permittivity?
Compare the results in Table.1 with literature.
Conclusions are too big in size. Make them more precise to attract the readers.
Author Response
Point 1: The title " High Permittivity Silicone Composites with Different Polarization Titanates for Electric Field Modification" clearly says high permittivity composites. Can you mention the frequency?
Response 1: Thank you for your comment. The frequencies at which the study was carried out, from 1 MHz to 25 Hz, relate to wide-spectrum equipment using different frequencies. This difference in the title is emphasized by the phrase "different polarization".
Point 2: How to modify the electric field to get high permittivity?
Response 2: Thank you for your comment. The use of materials with high permittivity, namely permittivity much greater than permittivity of the main insulation layer, modifies the electric field (based on this, the description of the Figure 9 is improved and explains in more detail the mechanism of modifying the electric field). (in red)
Point 3: The CaCO3 was used for synthesis. After reaction how can you confirm the absence of carbon dioxide?
Response 3: The chemical reaction for the synthesis of CCTO is as follows::
CaCO3 + 3CuO + 4TiO2 → CaCu3Ti4O12 + CO2↑
It occurs at temperatures not lower than 900 °C, and the decomposition of carbonate occurs in full. Confirmation of the obtained product is carried out by X-ray phase analysis of the obtained powder.
Point 4: Fig.1 is giving good graphical information for preparation. Provide some more discussion for this.
Response 4: Thank you for your comment. We have provided a more detailed discussion of this issue. (in red)
Point 5: For equation 1 & 2, give references.
Response 5: Thank you for your comment. References are given for equations 1 and 2. (in red)
Point 6: Fig.2 contains low resolution and i suggest comparing with JCPDS.
Response 6: We tried to improve the quality of Fig. 2.
Point 7: Determine the particle size for the samples using SEM.
Response 7: Particle size is discussed in the SEM analysis.
Point 8: Explain the reason for obtaining the shape of particles.
Response 8: The shape of the particles is determined by the parameters of the crystal lattice, but at smaller sizes, 3D structures tend to form shapes close to spherical. The ceramic powders used in this research were not subjected to high-energy milling, so they retained their shape according to the above-mentioned tendency.
Point 9: Fig.7 is for low frequencies. Are there any applications using this.
Response 9: For samples of silicone/CCTO/glycerin and silicone/KFTO/glycerin for frequencies below 10 kHz the dielectric loss tangent takes on such a large value that the practical application for such compositions is significantly limited.
Point 10: What is the difference between loss tangent and imaginary part of permittivity?
Response 10: The dielectric loss tangent is the ratio of the imaginary part of permittivity to the real one.
Point 11: Compare the results in Table.1 with literature.
Response 11: In the first column, literary sources are indicated in square brackets, i.e. a numerical comparison of the obtained results with literary data is carried out. The description in the text, discussion of Table 2 is additionally expanded. (in red)
Point 12: Conclusions are too big in size. Make them more precise to attract the read.
Response 12: Conclusions have been slightly shortened and presented in a more convenient form.
Reviewer 2 Report
Comments and Suggestions for Authors
The manuscript "High Permittivity Silicone Composites with Different Polarization Titanates for Electric Field Modification" reports composite silicone materials incorporating different fillers: CaCu3Ti4O12, KxFeyTi8-yO16, BaTiO3 and glycerol. The authors studied the electrical strength and permittivity and compared the results with those reported in literature.
I have the following observations about the results reported here:
Introduction:
-Why the authors have chosen glycerol as a plasticizer? The phase separation with silicone, high hydrophobic, must be discussed.
-The motivation of choosing silicones as matrices must be better emphasized. The limitations of silicones, the advantages with other matrices that the authors already have obtained and published the results.
-line 98-the authors claimed that glycerol is a good plasticizer for silicones- a reference must be added.
-lines 102-103 must be deleted, there is the same idea as lines 93-95.
Section 2: the silicone is missing in the Materials list. It must be added together with all the characteristics. Why this silicone is better than other? the advantages must be discussed.
-line 138-139 - the both components must be described, what kind of reaction occur betwen the components? Which are the polar groups to allow the compatibilization with glycerol? Why is necessary glycerol, as silicones form very flexible networks?
How the fillers were chemically compatibilized to allow a regular distribution in the matrix?
-How was established the suitable amount of filler in the matrix? based on which optimization protocols?
-Structural characterization of silicone is missing.
-There are some procedures to avoid the air in the mixture! The air pores also can originate from differences between the phases: silicone:glycerol!?
-Figs. 7 and 8: There is the same representation sign for two sets of samples and is difficult to understand.
-In Table 1 it is necessary to mention the molar mass of silicones.
I consider that the result prsented here must be compared with similar mixtures. A part of the silocones in the table are prepared in the laboratory and are not commercial available!
Based on these observations, I consider that the authors must clarify many weaknesses in the paper before acceptance!
Author Response
The manuscript "High Permittivity Silicone Composites with Different Polarization Titanates for Electric Field Modification" reports composite silicone materials incorporating different fillers: CaCu3Ti4O12, KxFeyTi8-yO16, BaTiO3 and glycerol. The authors studied the electrical strength and permittivity and compared the results with those reported in literature.
I have the following observations about the results reported here:
Introduction:
Point 1: Why the authors have chosen glycerol as a plasticizer? The phase separation with silicone, high hydrophobic, must be discussed.
Response 1: Thank you for your comment. References 32,33 and 34 have been added which discuss these issues in detail.
Point 2: The motivation of choosing silicones as matrices must be better emphasized. The limitations of silicones, the advantages with other matrices that the authors already have obtained and published the results.
Response 2: Thank you for your comment. As an additional motivation, the thesis on the Shore hardness of silicone and other polymer matrices studied earlier was added in the introduction.
Point 3: line 98-the authors claimed that glycerol is a good plasticizer for silicones- a reference must be added.
Response 3: References 32,33 and 34 have been added which discuss these issues in detail.
Point 4: lines 102-103 must be deleted, there is the same idea as lines 93-95.
Response 4: Thank you for your comment. It was deleted.
Point 5: Section 2: the silicone is missing in the Materials list. It must be added together with all the characteristics. Why this silicone is better than other? the advantages must be discussed.
Response 5: Relevant information on the silicone used has been added to section 2.
Point 6: line 138-139 - the both components must be described, what kind of reaction occur betwen the components? Which are the polar groups to allow the compatibilization with glycerol? Why is necessary glycerol, as silicones form very flexible networks?
Response 6: Relevant information on the motivation for the plasticiser used has been added to section 2.
Point 7: How the fillers were chemically compatibilized to allow a regular distribution in the matrix?
Response 7: Ceramic fillers are complex oxides that have low reactivity. The particle shape of these powders is favorable for good distribution in the polymer. Vacuum treatment minimizes the air that is formed during mixing.
Point 8: How was established the suitable amount of filler in the matrix? based on which optimization protocols?
Response 8: Thank you for your comment. The selection of the optimum amount of filler in the form of titanate powders was not the aim of the work. The percentage was chosen on the basis of literature data and remained unchanged for titanates with different polarisation types.
Point 9: Structural characterization of silicone is missing.
Response 9: Relevant information on the silicone used has been added to section 2 (Table 1).
Point 10: There are some procedures to avoid the air in the mixture! The air pores also can originate from differences between the phases: silicone:glycerol!?
Response 10: Thank you for your comment. We have provided a more detailed discussion of this issue. (Figure 2)
Point 11: Figs. 7 and 8: There is the same representation sign for two sets of samples and is difficult to understand.
Response 11: Thank you, corrected.
Point 12: In Table 1 it is necessary to mention the molar mass of silicones.
Response 12:
Point 13: I consider that the result prsented here must be compared with similar mixtures. A part of the silocones in the table are prepared in the laboratory and are not commercial available!
Response 13: In the first column, literary sources are indicated in square brackets, i.e. a numerical comparison of the obtained results with literary data is carried out. The description in the text, discussion of Table 2 is additionally expanded. (in red)
Round 2
Reviewer 2 Report
Comments and Suggestions for Authors
The authors provided valuable comments to support their reasearch and the paper can be accepted in the revised form.